# Identifying multi-resolution clusters of diseases in ten million patients with multimorbidity in primary care in England

Thomas Beaney [1,2] ✉, Jonathan Clarke [2], David Salman [1,3], Thomas Woodcock[1], Azeem Majeed[1], Paul Aylin [1] & Mauricio Barahona [2]

## Abstract

**Background** Identifying clusters of diseases may aid understanding of shared aetiology, management of co-morbidities, and the discovery of new disease associations. Our study aims to identify disease clusters using a large set of long-term conditions and comparing methods that use the co-occurrence of diseases versus methods that use the sequence of disease development in a person over time.

**Methods** We use electronic health records from over ten million people with multimorbidity registered to primary care in England. First, we extract data-driven representations of 212 diseases from patient records employing (i) co-occurrence-based methods and (ii) sequence-based natural language processing methods. Second, we apply the graph-based Markov Multiscale Community Detection (MMCD) to identify clusters based on disease similarity at multiple resolutions. We evaluate the representations and clusters using a clinically curated set of 253 known disease association pairs, and qualitatively assess the interpretability of the clusters.

**Results** Both co-occurrence and sequence-based algorithms generate interpretable disease representations, with the best performance from the skip-gram algorithm. MMCD outperforms k-means and hierarchical clustering in explaining known disease associations. We find that diseases display an almost-hierarchical structure across resolutions from closely to more loosely similar co-occurrence patterns and identify interpretable clusters corresponding to both established and novel patterns.

**Conclusions** Our method provides a tool for clustering diseases at different levels of resolution from co-occurrence patterns in high-dimensional electronic health records, which could be used to facilitate discovery of associations between diseases in the future.

## Plain language summary

Having multiple long-term conditions is linked to worse health, poorer quality of life, and difficulties accessing healthcare. Identifying groups, or 'clusters' of diseases that are more likely to occur together in one person may help healthcare services to better meet the needs of those with multiple conditions. Our study aims to identify clusters of similar diseases, based not only on the diseases someone has now, but on the order in which they developed them. We compare a range of methods and find that our strategy performs best at explaining diseases that are already known to be linked, whilst also identifying new clusters of diseases. These methods could be used in future to better understand how diseases occur together, which could help the design of more efficient healthcare services.

Multimorbidity, defined as the co-occurrence of two or more long-term conditions (LTCs) in one person, poses a significant challenge to health systems worldwide[1,2]. Having multimorbidity is associated with poorer quality of life[3], increased mortality[4], greater use of healthcare services, and higher healthcare costs[5,6]. As a binary label, multimorbidity is a crude marker of medical complexity[7] but there is growing evidence that distinct profiles or *clusters* of LTCs may be associated with differences in outcomes[8–10]. Although some clusters of co-occurring conditions are clinically well-established, for example, a cluster of conditions representing metabolic syndrome[11,12], the evolution of analysis methods for big data

opens up the use of routinely collected electronic health records (EHRs) for identifying clusters of less commonly occurring conditions. The anticipated benefits of the identification of disease clusters are summarised by Whitty and Watt (2020), as an opportunity "to uncover new mechanisms for disease; to develop treatments; and to reconfigure services to better meet patients' needs.[13]"

Over the last decade, many studies have been conducted to identify clusters of LTCs which co-occur together[14,15]. Among previous studies, mental health and cardio-metabolic conditions have consistently emerged as the two most replicable clusters[14,15]. However, current approaches to

[1]Department of Primary Care and Public Health, Imperial College London, London W6 8RP, UK. [2]Department of Mathematics, Imperial College London, London SW7 2AZ, UK. [3]MSk Lab, Department of Surgery and Cancer, Imperial College London, London W12 0BZ, UK. ✉e-mail: thomas.beaney@imperial.ac.uk

detect disease clusters suffer from limitations both in the use of data sources and in the approach to capture the multi-level complexity of disease associations. Firstly, most studies have used a relatively small number of LTCs (median=16 and range=10-99 for the 51 studies reviewed by Busija et al. (2019)) with coarse disease definitions (e.g., Diabetes, rather than considering separate subtypes). Secondly, most studies obtain only one clustering (with usually fewer than ten clusters), which may limit identification of associations between less common conditions[14]. As is the case with unsupervised methods, it is unlikely that there is one single configuration of clusters, but rather that a sequence of clusterings, from fine resolutions with many clusters to coarse resolutions with few clusters, may reveal more nuanced associations, and serve different purposes. Indeed, multiscale graph-based clustering methods, such as Markov Multiscale Community Detection (MMCD), enable the identification of clusters at different resolutions directly from the structure of the data, without the need to pre-specify the number of clusters or impose a hierarchical structure[16-18].

Recently, natural language processing (NLP) methods have emerged as a promising approach for handling the high-dimensional data found in EHRs[19-21]. When trained on word sequences in natural language, these predictive models learn a vector representation for each word, referred to as a word embedding, which captures semantic and syntactic characteristics of each word using the context in which it is used in text. In an analogous fashion, such models can be applied to the coded data in EHRs, where medical codes are words and EHRs are analogues to documents, to generate disease embeddings that capture information from their occurrence and the sequences observed in real data[19-23]. The disease embeddings can then be used to calculate the similarity between diseases for use in clustering. However, it remains unclear whether NLP methods incorporating additional information from the sequence of diseases recorded over time produce substantively different clusters to those obtained solely from co-occurrence-based methods, such as Multiple Correspondence Analysis (MCA), a dimensionality reduction method which has been used in several previous studies of multimorbidity clustering[24-26].

In this study, we aim to identify clusters of diseases from EHR data in an unsupervised manner, combining two recent approaches. Firstly, we generate disease representations applying two methods: one based only on co-occurrence (MCA), which is compared to newer NLP embedding methods that make use of code sequences. Secondly, we employ the multiscale graph-based clustering method of MMCD to identify disease clusters at different levels of resolution, based on the similarity of the obtained disease embeddings and compare against the k-means and hierarchical clustering algorithms commonly used in studies of disease clustering[14]. We apply these methods to a large and representative primary care EHR dataset of over 10 million patients in England and evaluate the resulting disease clusters to demonstrate that they provide clinically interpretable insights into disease associations.

## Methods

### Data sources and data cleaning

We used the Clinical Practice Research Datalink (CPRD) Aurum dataset, a nationally representative source of general practice (GP) data in England[27]. We included all patients aged 18 years or over, registered to a GP practice in CPRD Aurum between 1st January 2015 and 1st January 2020. Patients were censored at the earliest of date of deregistration, date of death, date of last collected data extraction from the practice, or the 1st January 2020. Any codes that were recorded or observed after the censoring date were excluded (see Supplementary Methods for details). Patients with two or more of the diseases defined below were included. Data cleaning rules for variables, including socio-demographics, are explained in detail in the Supplementary Methods.

### Disease definitions

Diagnostic codes are recorded in CPRD as Medcodes. These are entered by clinicians during clinical consultations and converted into a numeric code, for example, the term 'Allergic asthma' as Medcode ID 1483199016. We translated codes to a corresponding set of 212 LTCs. These were based on publicly available acute and chronic disease code-lists developed for the CALIBER study, where each list was developed by a panel of clinicians with expertise in the disease area[28]. Of the original 308 diseases, we used a subset of 211 conditions which were selected as representing chronic conditions by Head et al. (2021)[29]. As an example, the diagnostic codes representing 'Allergic asthma' and 'Exercise induced asthma' are grouped under the disease category of asthma. We reviewed the code-lists and supplemented them with an additional disease of chronic primary pain as a prevalent condition often included in multimorbidity studies (see Supplementary Methods)[30,31,32]. Diseases were ordered in sequences from earliest to latest according to timestamp of the observation, for example, a patient record might read sequentially as: "asthma, asthma, type 2 diabetes, hypertension, asthma, hypertension". We constructed two sequences for comparison: the first ("multiple") used all codes, and the second ("unique") included a disease only at its first occurrence (i.e., date of diagnosis); in the example above, this sequence is simplified as: "asthma, type 2 diabetes, hypertension". Where two codes had the same timestamp, we randomly ordered the corresponding codes.

### Generating disease embeddings

Figure 1 summarises the steps of our pipeline from data processing to clustering. We compared four different methods to create disease embeddings. As a baseline approach used previously in multimorbidity research, we used MCA[25,26]. Correspondence analysis (CA) is a class of methods which aim to reduce the dimensionality of binary or categorical data, analogous to Principal Component Analysis for continuous data, by minimising the chi-squared distance between observed and expected values based on the global co-occurrence matrix, or *Burt* matrix[33-35]. MCA is an extension of CA to two or more variables and has an advantage of allowing supplementary variables to be added which do not contribute to the calculation of the variance[34]. We applied MCA to the disease co-occurrence matrix, using the MCA algorithm implemented in Stata version 17.0 (StataCorp)[36].

We compared MCA to three popular NLP word embedding models: the word2vec models using continuous-bag-of-words (CBOW) and skip-gram (SG)[37], and Global Vectors (GloVe)[38]. CBOW and SG are related methods which use neural network architectures: in the case of CBOW, the model predicts a target word given a surrounding context window, whereas in SG, the model predicts the context given a target word[37]. In contrast, GloVe incorporates matrix factorisation of global co-occurrence statistics, combined with a local window[38]. Applied to text, these models are particularly effective at capturing semantic and syntactic word analogies, for example, capturing the relationship of 'king is to queen as man is to woman'[38]. In each case, we compared the default hyperparameter values of the models to values we hypothesised might better represent the smaller vocabulary and relatively short sequences (in comparison to the documents for which the methods were originally developed). We then selected the best performing model according to our evaluation metrics below.

For CBOW and SG, we used the word2vec model implemented in the *gensim* package for training on the sequences of all 10.5 million patients[39]. We compared vector sizes of 10 and 30, window sizes of 2 and 5, negative sampling of 2 and 5, and down-sampling of frequent diseases comparing the default of 0.001 to no down-sampling. For GloVe, we used the *glove-python* implementation and compared a default window size of 5 to values of 2, and learning rate of 0.05 to values of 0.01 and 0.1[40]. We also tested models over a range of epochs as detailed in the Supplementary Methods.

### Evaluation of embeddings

We evaluated our embedding methods using a curated set of 253 known disease association pairs. These were created by three co-authors with a clinical background, TB, JC, and DS, based on the 212 available diseases, using the British Medical Journal Best Practice guidelines and clinical judgement as detailed in the Supplementary Methods[41]. For each embedding model we proceeded as follows: for each disease $d_1$ in the set of known

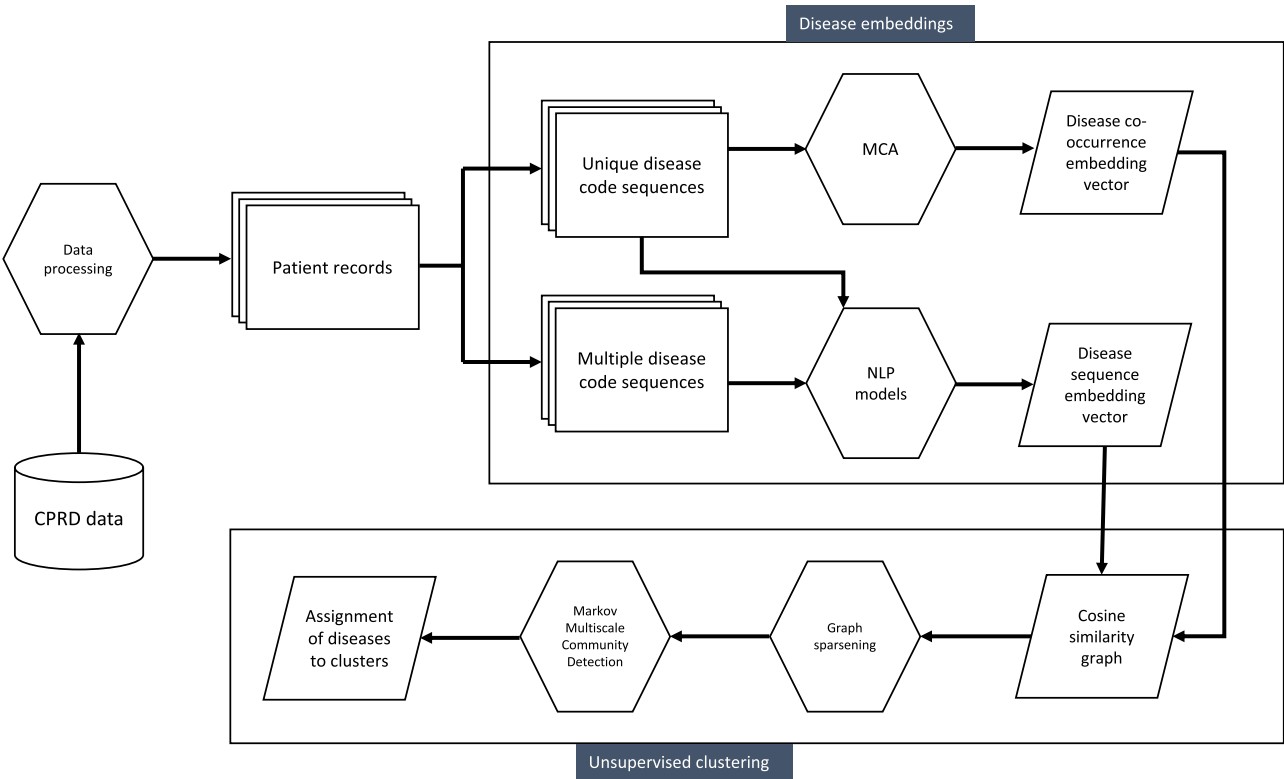

**Fig. 1 |** Pipeline for generating disease clusters from Clinical Practice Research Datalink (CPRD) data.

disease association pairs, we calculated the percentage of known associated diseases ($d_2 \dots d_N$) that were in the set of ten most similar diseases of $d_1$ in terms of cosine similarity computed from the embedding. Similar approaches have been used by other authors, with Solares et al. (2021) using a range of neighbourhood sizes from three to 20[21]. Beam et al. (2019) used bootstrap sampling of the similarity distribution for each condition, and assigned conditions if in the top 5% of the distribution of most similar conditions, which is roughly equivalent to use of the top ten conditions in our case (given 212 conditions)[23]. We checked the robustness of our evaluation by comparing different thresholds of neighbourhood sizes of two, five, and 20.

### Markov Multiscale Community Detection
To cluster the selected disease embeddings, we used MMCD. The first step is to construct a similarity graph of diseases, a sparsified weighted graph where the diseases are the nodes of the graph and the weights represent the similarity between the embedding vectors. To construct the graph from the data, we calculated the pairwise cosine similarity matrix $S$ for all diseases and followed the normalisation approach of Altuncu et al. (2019), by calculating the distance matrix $D = 1 - S$, applying max normalisation to give $\hat{D}$, and then calculating the normalised cosine similarity as $\hat{S} = 1 - \hat{D}$[42]. This produces a dense similarity matrix, which is then sparsified to transform it into a *similarity graph*. This sparsification is a key pre-processing step in MMCD as it removes edges with weaker associations. Although simple thresholding based on weights was originally applied in this step, it is not robust to noise and does not capture well the inhomogeneities in the similarities in the data. Hence, several methods have been proposed for this step using global constructions that involve the minimum spanning tree (MST), which contains the collection of edges with minimum weight sum that fully connect all nodes on the graph, thus ensuring global connectivity. To this sparse network, edges representing local connectivity are often added, such as the k-nearest neighbours (kNN) for each node. Recently, Liu and Barahona (2020) demonstrated improvement on the kNN graph by using continuous kNN (CkNN)[18]. In CkNN, for distance $d_{i,j}$ connecting nodes $i$ and $j$, and where $d^k(i)$ and $d^k(j)$ are the distances to the $k$-th nearest

neighbour of $i$ and $j$, respectively, then the edge is retained if:

$$d(i,j) < \partial \sqrt{d^k(i) d^k(j)} \qquad (1)$$

where $\partial$ is a parameter that can be varied to alter the sparsity of the network. In our case, we hold $\partial$ constant at a value of 1, but vary the value of $k$; as shown by Liu and Barahona (2020), the MMCD algorithm is relatively robust to different parameterisations[18]. We selected a CkNN value of ten, but comparison of CkNN values of five, fifteen and twenty resulted in similar partitions.

To this sparsened undirected network, we then applied MMCD, using the *pygenstability* module in Python[17,43,44] which models a random-walk across the network, and evaluates subgraphs of the original graph over which the Markov dynamics is contained over a time $t$ that acts as a scale. The natural scanning over scales performed by the diffusion on the graph reveals larger communities (i.e., coarser clusterings) as the scale $t$ is increased[17]. For details of the method and its applications see Delvenne et al. (2010) and Arnaudon et al. (2021)[17,43]. At each time step, we optimise the cost function 2000 times using the Leiden algorithm[45], and calculate the normalised variation of information (NVI), an information theoretic measure for comparing cluster partitions, where 0 indicates identical partitions and 1 indicates dissimilar partitions[46]. The algorithm then selects partitions that have low values of the NVI across scales and also with respect to the Leiden optimisation, using the automated scale selection algorithm developed by Schindler et al. (2023) which smooths the NVI to identify persistence across scales[47]. Models were run over a Markov scale aiming for between 4 and 30 clusters, using 500 scale steps, 2000 optimisation evaluations and select 400 optimisations to compute the NVI at each scale[44].

### Benchmarking and evaluation of clusters
As a benchmark, we compared the cluster partitions derived from MMCD to k-means and hierarchical clustering using Ward's method, as baselines widely used in multimorbidity research[14]. We compared to the same

number of clusters as identified in MMCD. In contrast to MMCD, which used the cosine similarity matrix, these methods were applied directly to the disease embeddings as input features. We implemented k-means with the Lloyd algorithm, iterating 1000 times with different random centroid seeds. We used *scikit-learn* package for both k-means and Ward's clustering algorithms[48].

To enable comparison across methods, we calculated metrics related to the interpretability of clusters. As an intrinsic measure of the relevance of disease clusters to patterns of diseases in patients, we first randomly sampled 100,000 patients with replacement. For each patient, we then randomly sampled two different diseases from their set of all diseases, once per patient. We assigned patients to a disease cluster if *both* diseases were contained within the same cluster. Of these patients $\{P_1, \dots P_N\}$ assigned to a disease cluster, we calculated a metric of the pairwise Jaccard similarity between the set of two diseases $\{d_1, d_2\}$ for each patient in the same cluster, and report the arithmetic mean of all possible pairs.

To compare between the three clustering algorithms for partitions with the same number of clusters, we used information from the 253 known disease pairs. We expect that in a more interpretable clustering solution, known disease pairs are more likely to be assigned to the same cluster. To correct for the bias that favours unbalanced and uninformative clustering solutions with all diseases assigned to a single cluster, we considered the observed assignment of known disease pair edges within clusters to that expected, assuming the contingency table in Table 1.

Following from Table 1, we calculated the odds ratio for a known disease pair edge being intra-cluster compared to inter-cluster:

$$OR = \frac{DP_{intra} \times E_{inter}}{DP_{inter} \times E_{intra}} \quad (2)$$

A higher OR here can be interpreted as higher odds that a known disease pair edge is found in the same cluster given the cluster distribution for a partition with the given number of clusters, representative of more balanced and informative clusters.

### Comparison to ICD-10 classification

We compared the clusters to the system a condition is assigned to in the CALIBER code-lists, which is corresponds closely to the classification of chapters in ICD-10[28], using the normalised variation of information (NVI). Each disease is assigned to one of sixteen systems, for example, asthma is assigned under 'Diseases of the Respiratory System', similar to the chapters used in ICD-10.

We used Python version 3.8.10 and Pandas version 1.3.5 for data manipulation and management[49,50]. Sankey diagrams were created in Plotly[51].

### Reporting summary

Further information on research design is available in the Nature Portfolio Reporting Summary linked to this article.

## Results

### Description of the data

Of 15,256,726 patients aged 18 years or older registered in CPRD Aurum in England from 1st January 2015 to 1st January 2020, there were 10,579,232 (69.3%) with at least two of a pre-defined set of 212 LTCs and were thus included in the study. Characteristics of the eligible cohort are displayed in Supplementary Table 1. The median age was 52 (IQR: 36–68) years. There

were more females than males (53.4% vs 46.6%) with a small number (263) recorded in CPRD as "indeterminate" gender. The majority (73.0%) of people were recorded as being of White ethnicity, with 13.9% having no recorded data on ethnicity. There was a roughly even split between deciles of socioeconomic deprivation (measured by the Index of Multiple Deprivation), but with relatively fewer in the most deprived decile (9.1%). For each patient, we constructed two sequences for comparison: the first (multiple) used all diagnostic codes representing the 212 LTCs, and the second (unique) included a code only at its first occurrence. Using the unique code sequences, the median number of codes per patient was 5 (IQR: 3–9); using multiple code sequences, the median was 13 (IQR: 6–33) (Supplementary Table 1 and Supplementary Fig 1). Raised total cholesterol had the highest code occurrence of unique code sequences (5,408,007) and hypertension had the highest code occurrence including multiple code sequences (29,299,147) (Supplementary Table 2).

### Disease embeddings

We generated two different disease embeddings (see Fig. 1). Applying MCA, as shown by the scree plot (Supplementary Fig 2), the first two dimensions explained a large amount of the variance: 58.3% for the first, and 5.5% for the second. As expected, the first dimension largely reflected increasing age and number of conditions (see Supplementary Fig 3). To evaluate our embeddings, we developed a set of 253 clinically well-established disease pairs. Using this set of disease pairs, 30 dimensions from MCA resulted in the optimal number of disease pairs being assigned in the top ten nearest neighbours to each disease based on the cosine similarity calculated from the MCA embeddings (see Methods and Fig. 2).

We next generated embeddings using three NLP models: CBOW, SG, and GloVe, trained on each of the unique and multiple code sequences from all 10.5 million patients. We tested a range of hyperparameter values, and the optimal hyperparameters were chosen for each of the three NLP models using the same evaluation strategy as for MCA (see Supplementary Tables 3–6). When evaluated against the curated set of 253 known disease pairs, GloVe and SG had similar performance to MCA-30 for unique code sequences, with lower performance for CBOW (Fig. 2). The NLP models had comparatively better performance when run on multiple code sequences versus unique code sequences, indicating that additional information is provided by the sequence of reappearing codes. In a sensitivity analysis, model performance was similar when comparing against the nearest two, five or twenty neighbours (Supplementary Fig 4). Overall, SG with multiple codes (SG-M) showed the best performance across all models. We thus selected SG-M with an embedding dimension of 30 as the best-performing NLP embedding and compared it to the best co-occurrence embedding (MCA-30) for clustering.

### Clustering of disease embeddings

We applied a multiscale clustering algorithm (MMCD) to the cosine similarity between both disease embeddings (MCA-30 and SG-M). Using the MCA-30 embeddings, MMCD identified optimal clusterings at three resolutions representing 23, nine, and six clusters (Fig. 3). Using the SG-M embeddings, optimal clusterings were identified at 25, fifteen, seven, and five clusters and we selected the first three of these for further evaluation (Fig. 4). In both cases, Sankey diagrams demonstrated that most conditions in a cluster remained in the same cluster across levels of resolution (Figs. 5 and 6). This indicates a quasi-hierarchical pattern of similarity between diseases, with smaller groups of diseases showing greater similarity and, in turn, getting integrated into broader disease groups with a looser observational similarity.

### Comparison to other clustering methods

To evaluate our clustering method, we compared the clusters derived from MMCD to two clustering algorithms widely used in studies of disease clustering: k-means and Ward's hierarchical clustering[14]. In each case, we selected the same corresponding number of clusters to those from MMCD. For both embedding methods, k-means, and hierarchical clustering

**Table 1 | contingency table of assignment of known disease pairs to clusters**

| | Intra-cluster | Inter-cluster |
|---|---|---|
| Known disease pair edges | $DP_{intra}$ | $DP_{inter}$ |
| Other disease pair edges | $E_{intra}$ | $E_{inter}$ |

**Fig. 2 | Percentage of disease associations from a curated set of 253 known disease pairs that are assigned to the ten nearest neighbours based on cosine similarity for each disease embedding.** A data table with the exact values underlying the figure is given in Supplementary Table 9.

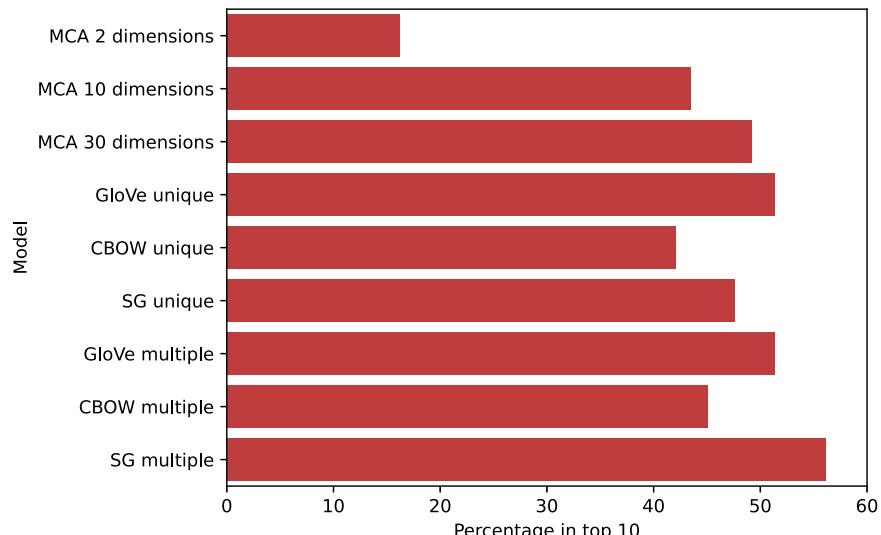

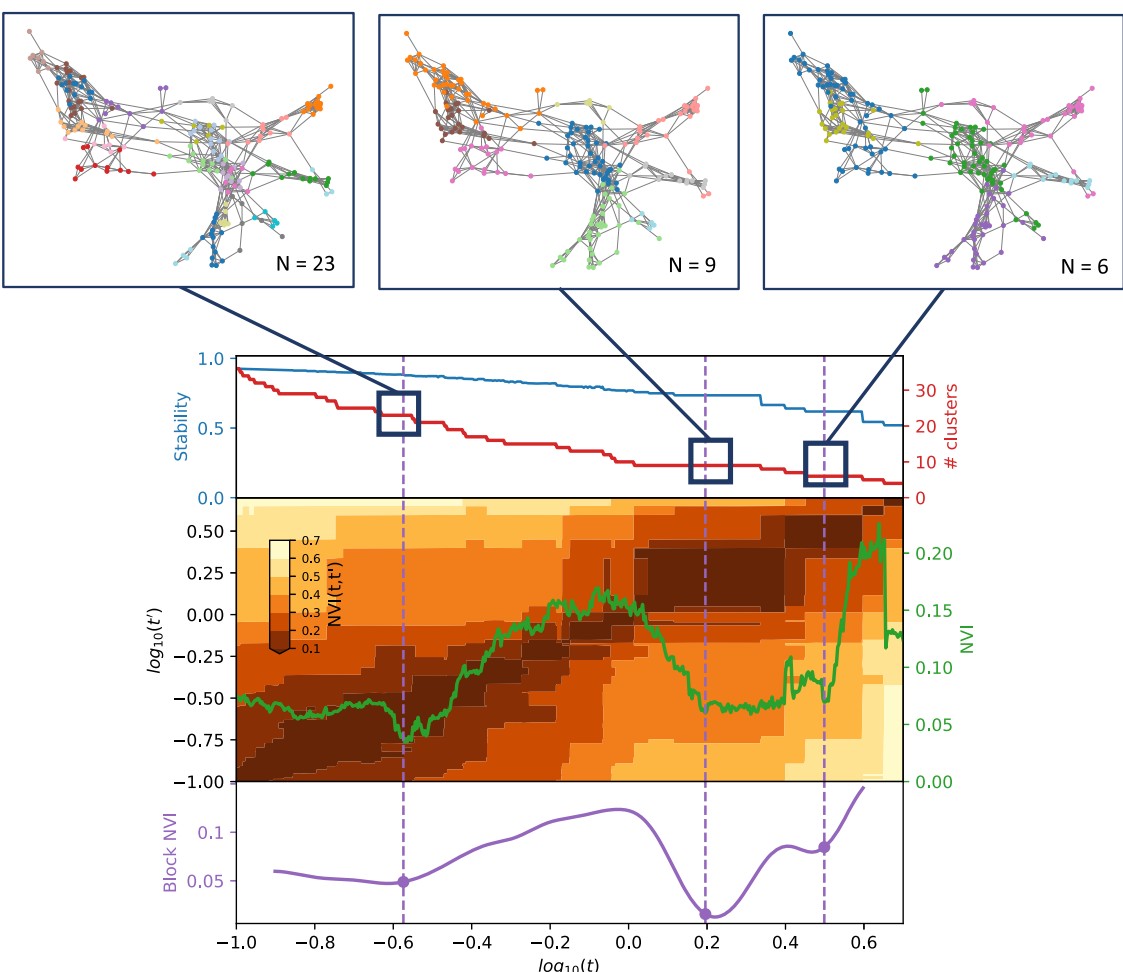

**Fig. 3 | Selection of optimal clusterings from Markov Multiscale Community Detection using MCA-30 embeddings.** The optimal clusterings at 23, nine and six clusters. The disease similarity graphs obtained with CkNN for the three optimal clusterings are shown above, where the nodes correspond to diseases, coloured by cluster assignment, and edges to strong similarities. In the trace below, the shaded areas correspond to partitions across scales, where darker areas correspond to more robust partitions. The NVI (green line) represents the variation in the assignment of diseases to clusters within each Markov time step, $t$, and the purple line represents the block NVI across $t$; minima of these traces represent robustness within and across scales, respectively (see Methods). MCA-30 = Multiple Correspondence Analysis retaining 30 dimensions.

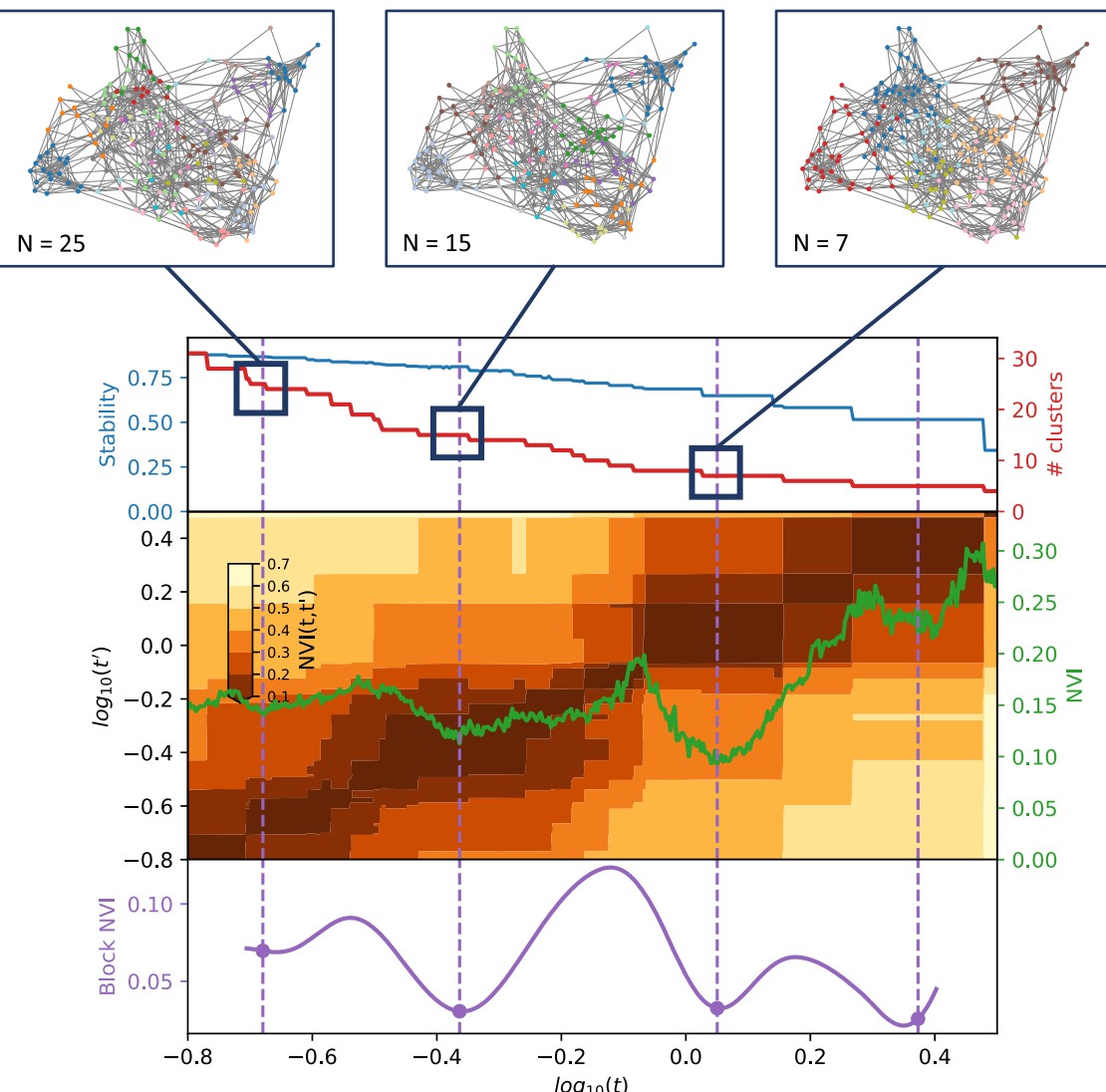

**Fig. 4 | Selection of optimal partitions from Markov Multiscale Community Detection using SG-M embeddings.** The optimal clusterings contain 25, fifteen seven, and five disease clusters and we focus on the displayed clusterings with 25, fifteen, and seven disease clusters. The disease similarity graphs obtained with CkNN for the three optimal clusterings are shown above, where the nodes correspond to diseases, coloured by cluster assignment, and edges to strong similarities. In the trace below, the shaded areas correspond to partitions across scales, where darker areas correspond to more robust partitions. The the NVI (green line) represents the variation in the assignment of diseases to clusters within each Markov time step, $t$, and the purple line represents the block NVI across $t$; minima of these traces represent robustness within and across scales, respectively (see Methods). SG-M = Skip-Gram using Multiple code sequences.

produced unbalanced partitions, with a few dominant clusters and some additional very small clusters containing few diseases. Using our curated set of 253 clinically established disease associations, we found that known disease pairs were substantially more likely to be assigned to the same cluster using MMCD (Fig. 7). Furthermore, although randomly sampling any two diseases from one patient, a patient was more likely to have both conditions assigned to the same disease cluster using hierarchical and k-means clustering, due to the large size of the dominant clusters, they were less likely to share conditions with other people in the same cluster (Supplementary Fig 5) across the range of partitions.

### Comparison of clusters to ICD-10 chapters

We also compared the MMCD disease clusters to the assignment of the diseases in the corresponding sixteen chapters of the ICD-10 medical taxonomy by computing the NVI, where NVI = 0 indicates perfect agreement and NVI = 1 corresponds to maximum disagreement. With the MCA-30 embeddings, the similarity to the ICD-10 chapters ranged from NVI = 0.60

for 23 clusters to NVI = 0.75 for six clusters (Supplementary Table 7A). With the SG-M embeddings, the NVI was slightly lower than that for MCA-30, ranging from 0.55 for 25 clusters to 0.68 for seven clusters (Supplementary Table 7B). These results indicate a substantial mismatch in the groupings of diseases within the MMCD clusters compared to the ICD-10 chapters, reflecting the difference between data-driven co-occurrence patterns and a clinical taxonomy.

### Descriptive evaluation of clusters

Given its higher performance, we considered only the MMCD clusterings for further descriptive evaluation. To aid visualisation and interpretation, clusters were assigned a descriptive label aiming to represent most of the diseases in the cluster. Figures 5 and 6 show Sankey diagrams capturing the quasi-hierarchical organisation of the MMCD clusters obtained for both MCA-30 and SG-M embeddings, whereas Figs. 8 and 9 provide a more detailed visualisation of the contents of the disease clusters across resolutions.

**Fig. 5 | Sankey diagram of clusters at resolutions of 23, 9, and 6 clusters, using MCA-30 embeddings.** Clusters within a single partition are represented by nodes of the same colour. Lines connecting nodes of different colours are weighted according to the number of conditions in each cluster and represent the number of conditions that are in the corresponding cluster at a coarser resolution. CKD = Chronic Kidney Disease; HF = Heart Failure; LD = Learning Disabilities; MH = Mental Health; MSK = Musculoskeletal; MCA-30 = Multiple Correspondence Analysis retaining 30 dimensions.

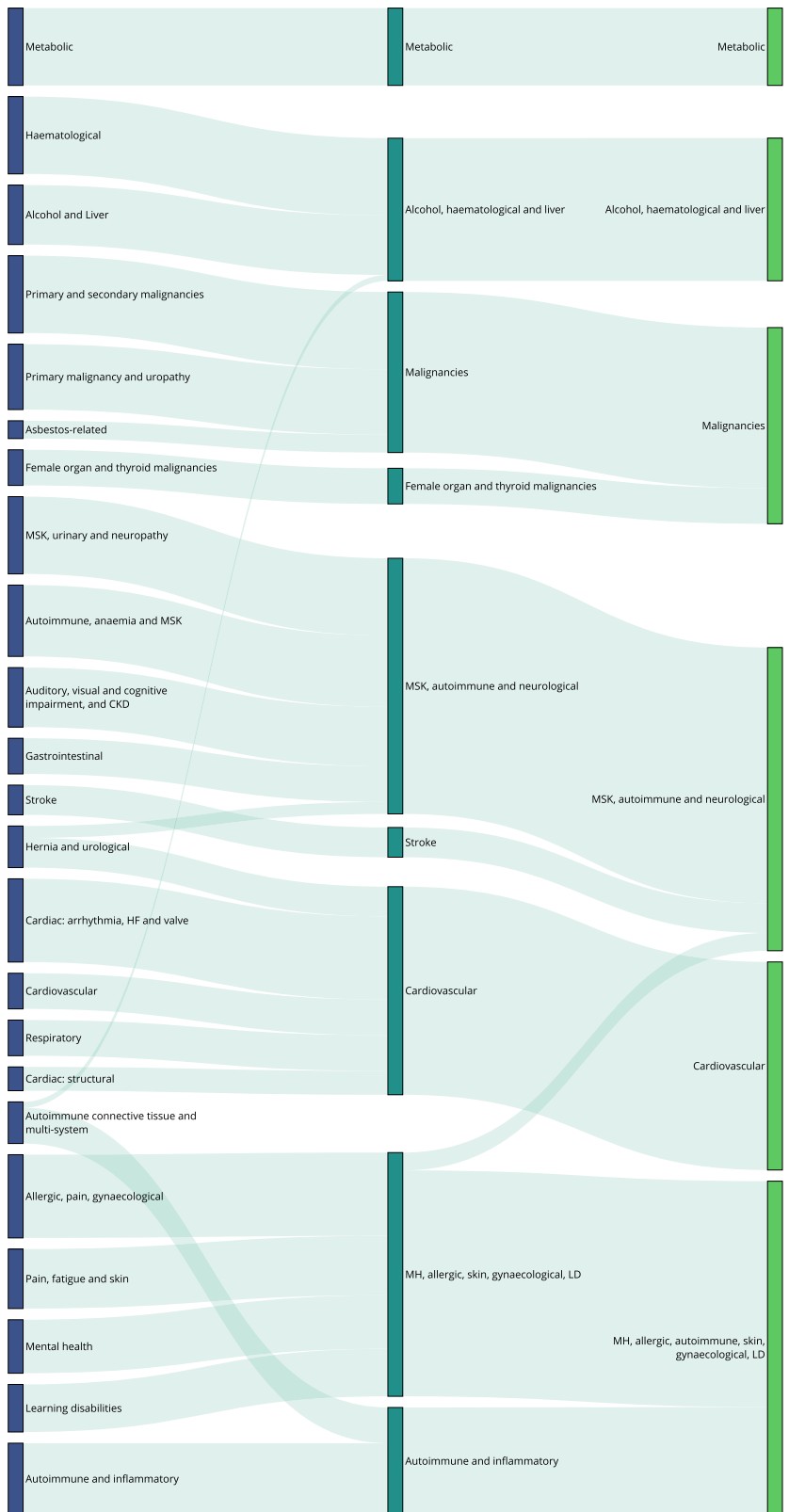

## Clusters from MCA-30 embeddings

At the 23-cluster resolution, several well-defined clusters were identified, including a cluster representing the established metabolic syndrome[52] (including obesity, raised cholesterol, hypertension, diabetes, and diabetes complications), forms of stroke, autoimmune and inflammatory conditions, and liver conditions (Fig. 8). Many malignancies clustered together at this fine resolution, except for breast, gynaecological and thyroid primary malignancies, which clustered separately, and primary malignancy of the skin and prostate, which clustered separately along with urological conditions. As would be expected from similarities drawn from co-occurrences in data, some clusters reflected diseases common in particular age groups, for example a cluster of diseases affecting younger

**Fig. 6 | Sankey diagram of clusters at resolutions of 25, 15, and 7 clusters, using SG-M embeddings.** Clusters within a single partition are represented by nodes of the same colour. Lines connecting nodes of different colours are weighted according to the number of conditions in each cluster and represent the number of conditions that are in the corresponding cluster at a coarser resolution. GI = Gastrointestinal; HF = Heart Failure; LD = Learning Disabilities; MH = Mental Health; MSK = Musculoskeletal; SG-M = Skip-Gram using Multiple code sequences.

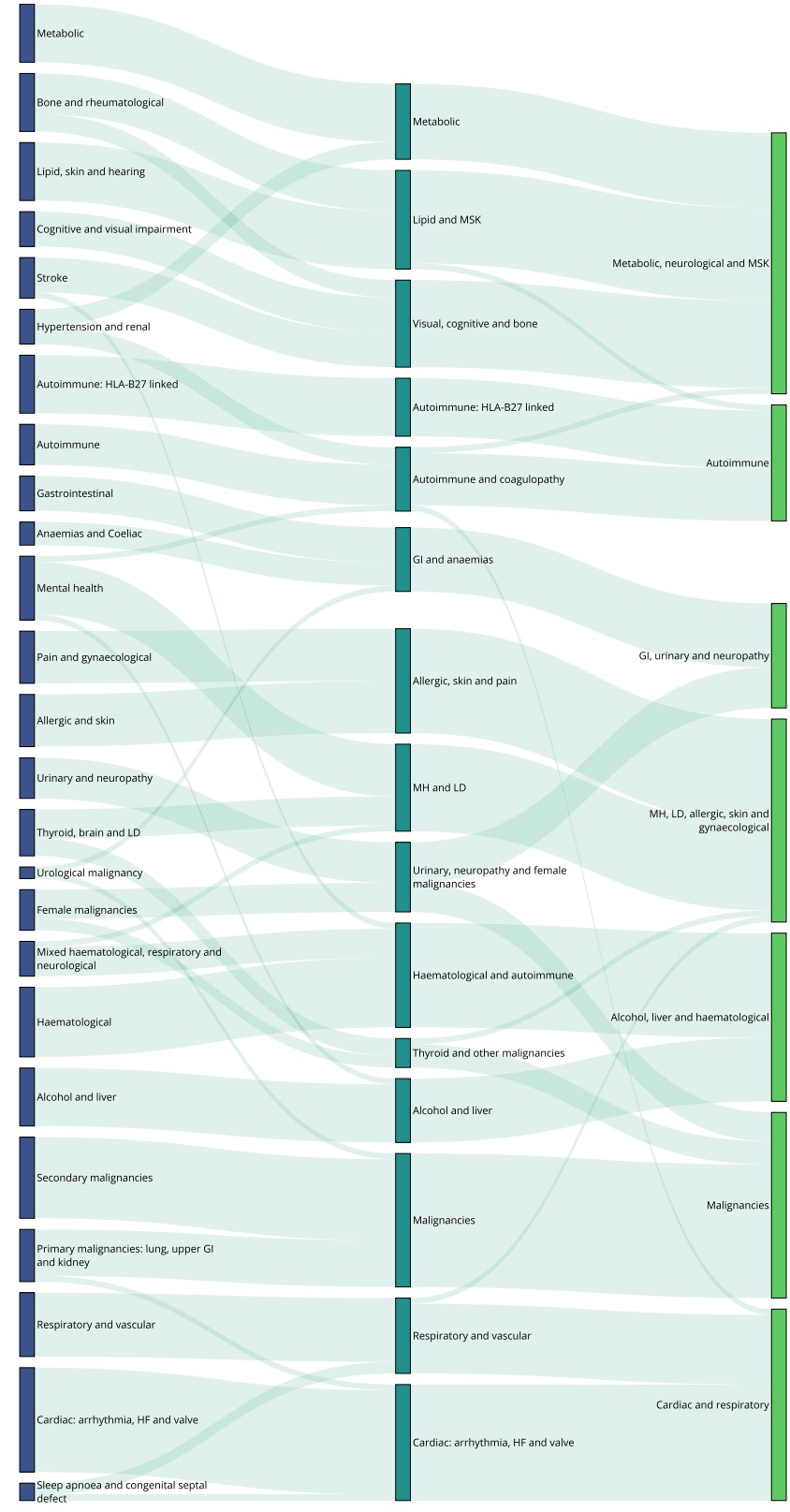

people (including acne, dysmenorrhoea, polycystic ovarian syndrome, and allergic and chronic rhinitis), and another cluster with diseases more common in older people (including dementia, hearing loss and visual impairment).

The sequence of clusterings at multiple resolutions revealed that most of the disease clusters at the 23-cluster resolution integrate quasi-

hierarchically at coarser resolutions (Fig. 5). Notably, the metabolic cluster displayed the strongest stability, with the same conditions clustered together across all resolutions. However, some diseases separated from their assigned clusters across different scales. For example, cystic fibrosis (CF) was present in an 'Autoimmune and inflammatory' cluster at a resolution of 23 clusters but joined an 'Alcohol, haematological and liver' cluster at coarser

**Fig. 7 | Odds ratios for assigning a known disease pair to the same cluster compared to the expected distribution of 253 known disease pairs. A** displays results using the MCA-30 embeddings and (**B**) displays results using the SG-M embeddings. Data tables with the exact values underlying the figure are given in Supplementary Tables 10 and 11.

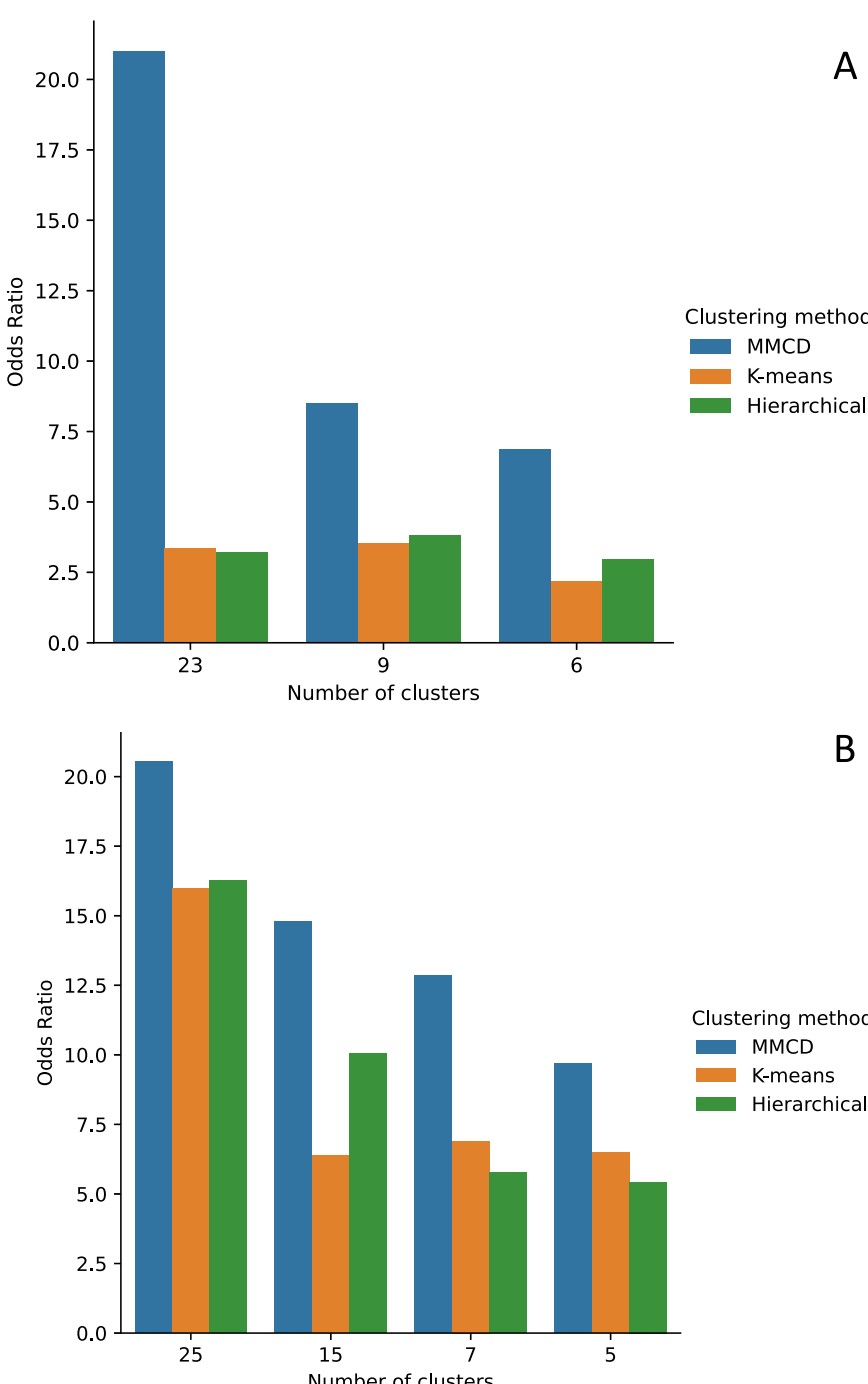

resolutions, which may reflect the challenge of assigning a multi-system disease such as CF to a consistent set of clusters.

### Clusters from SG-M embeddings

The clusters derived from the SG-M embeddings, which consider not only co-occurrence patterns but also information contained in the multiple code sequence, were different to those from MCA-30. At the fine 25-cluster resolution, well-defined clusters include, for example, one representing stroke sub-types and another representing heart failure, valvular, and arrhythmogenic cardiac conditions (Fig. 9). As with MCA-30, a metabolic cluster including diabetes and obesity was observed, but hypertension was clustered instead with renal diseases, and both Raised Total Cholesterol and Raised LDL-C clustered separately with enthesopathy, hearing loss, and skin cancer. We found instances of clustering

according to underlying causal mechanisms: two separate autoimmune clusters are present in this clustering, one including rheumatoid arthritis and related diseases, and another including spondyloarthropathies and inflammatory bowel disease which are strongly associated with the HLA-B27 gene[53].

The quasi-hierarchy of the partitions across resolutions in SG-M is less strong than for MCA-30 (Fig. 6), reflecting the additional complexities contained in the contextual information of sequences captured by NLP embeddings. For example, thyroid cancer was clustered with thyroid disease and learning disabilities at the 25-cluster resolution, which could be partially attributed to a well-established link between thyroid disease and Down's syndrome[54]. At the fifteen-cluster resolution, thyroid disease and thyroid cancer joined a cluster with melanoma, testicular, and brain cancer. While people with melanoma may be at higher risk of developing thyroid

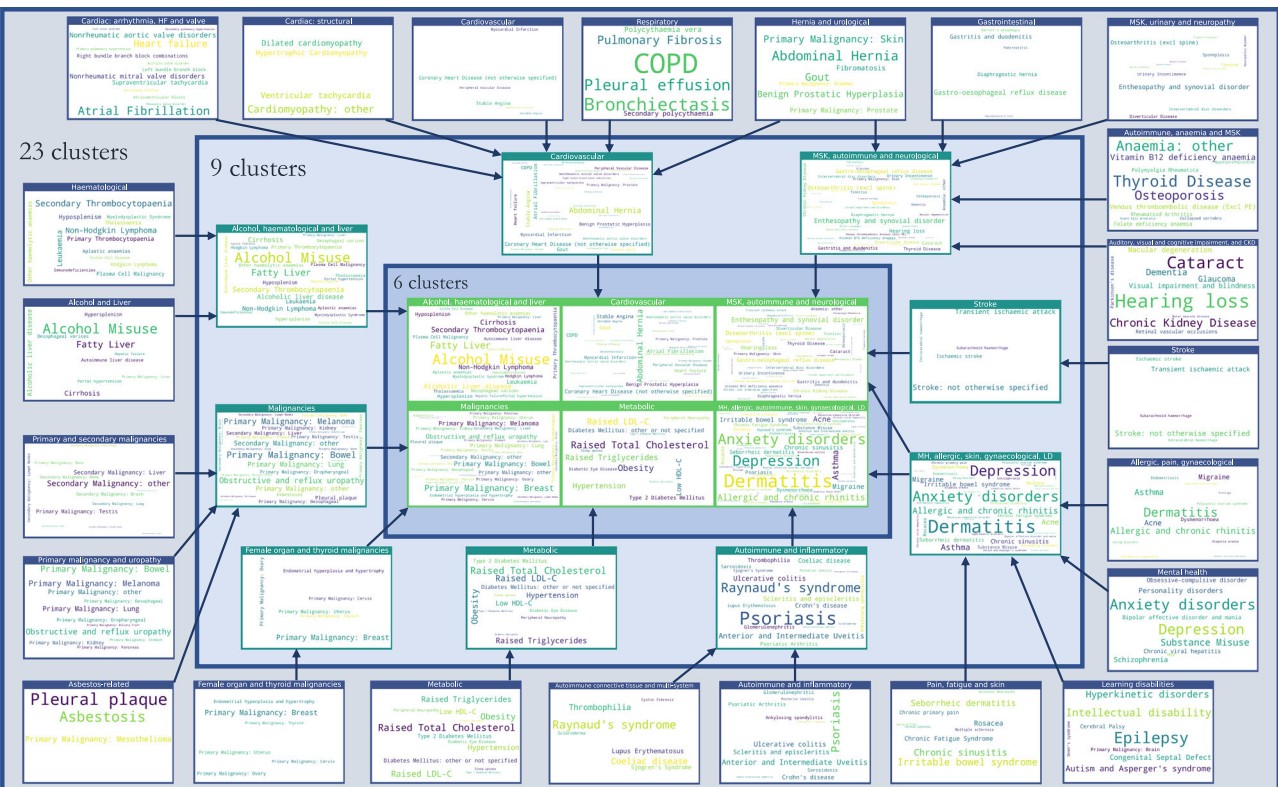

**Fig. 8 | Assignment of diseases to clusters at resolutions of 23, 9, and 6 clusters, using MCA-30 embeddings.** Arrows assigned between clusters if at least two conditions, or ≥20% of conditions within a cluster are assigned to a cluster at a coarser resolution. CKD = Chronic Kidney Disease; HF = Heart Failure; LD = Learning Disabilities; MH = Mental Health; MSK = Musculoskeletal; MCA-30 = Multiple Correspondence Analysis retaining 30 dimensions.

malignancy[55], other associations within this cluster were unexpected. To investigate this cluster further, we compared the observed to expected ratio of co-occurrence for each pair of conditions in the cluster (Supplementary Table 8). This demonstrated a stronger than expected ratio of co-occurrence of thyroid disease and thyroid malignancy (9.29), of primary brain cancer with melanoma (2.29), and of thyroid cancer with melanoma (2.35). Testicular cancer had a lower ratio of co-occurrence with thyroid disease and thyroid cancer (0.45 and 0.89, respectively), but higher-than-expected ratio of co-occurrence with brain cancer and melanoma (1.96 and 1.17, respectively), demonstrating that appearance together in a cluster needs to be examined in more detail, as it does not necessarily indicate that each disease is directly associated with every other disease in the cluster.

## Discussion

Our study presents an application of an unsupervised, multiscale graph-based clustering method (MMCD) to vector embeddings of diseases derived from EHR data of 10.5 million patient records. Our analysis produces interpretable clusters of diseases from fine to coarse resolutions, based on the intrinsic patterns of co-occurrence and sequences of diseases in people with multimorbidity. We found that MMCD outperformed k-means and hierarchical algorithms in clustering pairs of diseases known to be associated using disease embeddings generated from both co-occurrence-based MCA and sequence-based NLP methods. We also find optimal clusterings over multiple resolutions, highlighting the advantages of considering a range of levels of coarseness. Although a full description of the relationships of all 212 diseases was outside the scope of this study, we demonstrate the power of these methods for classifying multimorbidity clusters at different resolutions, which may help identify more fine-grained relationships in future research. We also provide access to the disease embeddings and cluster assignments, as an open resource for other researchers.

Clusters derived from MCA-30 and SG-M embeddings differed, but both pick up meaningful patterns of diseases that are clinically interpretable.

In general, clusters from SG-M were less interpretable than those from MCA-30, which is likely to reflect the additional contextual information captured in disease sequences, beyond those captured by co-occurrence alone. It may also reflect differences in coding frequency between diseases, with previous work indicating that some diseases are more likely to have recurrent codes, particularly those with financial incentives attached to their management[56]. Although there is no gold-standard ground truth for disease clusters to be compared to, conditions that are known to form part of the well-established metabolic syndrome[52] clustered together across resolutions in MCA-30, while other clusters represented conditions with similar underlying causal mechanisms, for example, those associated with the HLA-B27 gene[53]. In contrast, other clusters represented LTCs that may occur at similar ages, for example, clusters including dementia and cataracts from both MCA and SG-M derived embeddings, conditions which are more prevalent in older people. The clusterings may therefore capture different factors explaining disease co-occurrence, related to genetics, demographics, or direct causal relationships.

Finer resolutions with more clusters are likely to be the most valuable in identifying novel disease associations and, at these resolutions, we found more unexpected disease patterns that suggest avenues for further investigation, for example, the grouping of thyroid cancer, thyroid disease, melanoma, testicular and brain cancer in SG-M. However, as we found, the cluster assignment does not necessarily indicate that each disease is directly associated with every other disease (Supplementary Table 8). This may in part be due to a distinguishing feature of sequence-based models compared to co-occurrence-based models, whereby the representation is determined not only by direct co-occurrence but by shared associations of two diseases with other diseases, capturing indirect information which is less obvious.

The clusters demonstrate a remarkably hierarchical structure with MCA-30, and to a lesser extent, with SG-M. This is an intrinsic feature of the data, rather than MMCD, which does not impose any hierarchical structure on the sequence of clusterings. Our findings suggest that using hierarchical

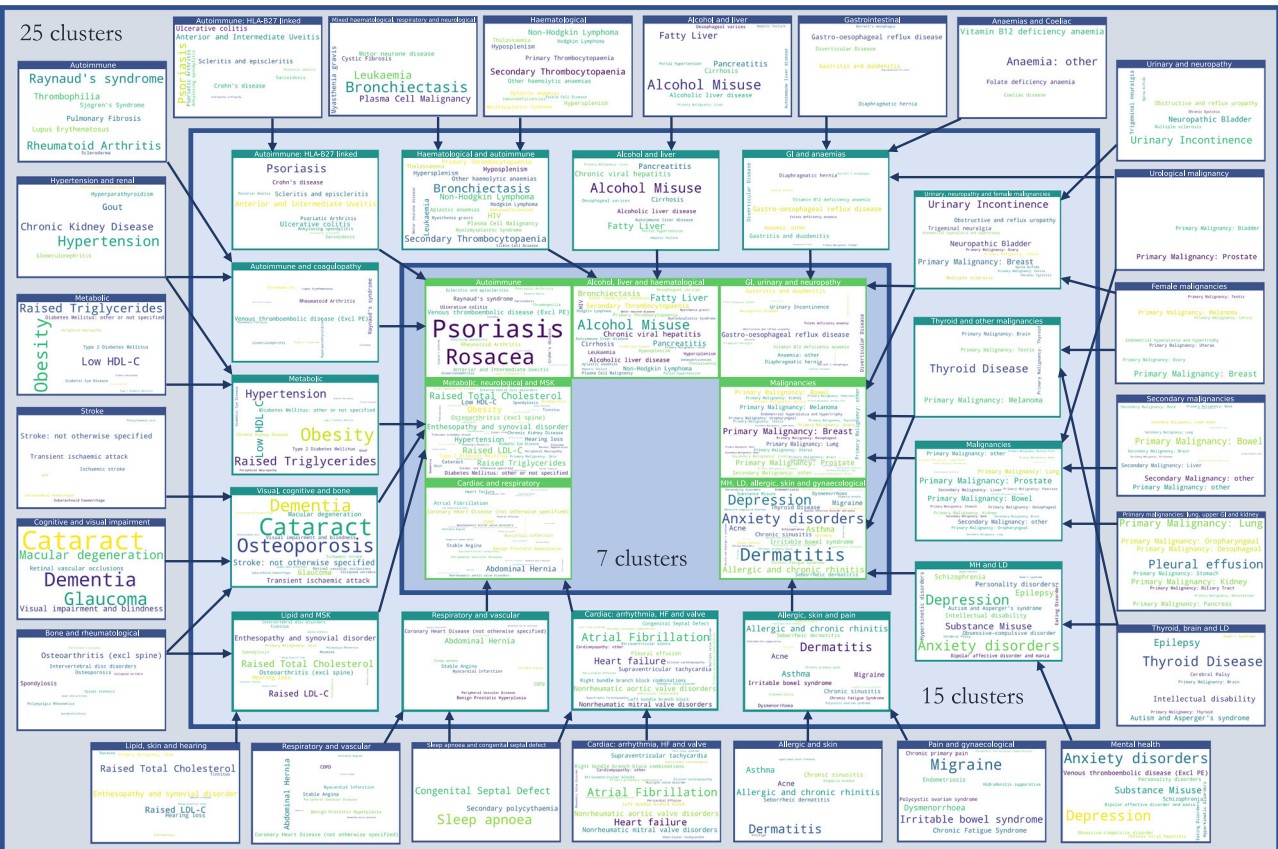

**Fig. 9 | Assignment of diseases to clusters at resolutions of 25, 15, and 7 clusters, using SG-M embeddings.** Arrows assigned between clusters if at least two conditions, or ≥20% of conditions within a cluster are assigned to a cluster at a coarser resolution. CKD = Chronic Kidney Disease; HF = Heart Failure; LD = Learning Disabilities; MH = Mental Health; MSK = Musculoskeletal; SG-M = Skip-Gram using Multiple code sequences.

clustering algorithms that enforce a hierarchical structure may mask meaningful variation in the structure of the data over different resolutions. For example, in both MCA-30 and SG-M embeddings, CF appeared with different diseases across resolutions. As a condition linked to both liver disease[57] and higher prevalence of anxiety and depression[58], its separation at different resolutions likely represents the challenge of assigning a multi-system disease to a single branch of hierarchical clusters. Our multi-resolution approach thus provides the advantage of allowing assessment of the stability of disease assignment to clusters across resolutions as a means of drawing further information.

Some studies have evaluated the quality of embeddings by comparing clusters to known hierarchical disease taxonomies, such as ICD-10, which are predominantly based around organ systems[20,21]. We found that our disease clusters differed substantially to the classification of the ICD-10 chapters, highlighting the disparity with systems-based classifications and suggesting that hierarchical taxonomies are not a suitable method by which to evaluate the quality of disease similarity based on co-occurrence or sequence.

A strength of our work is the direct comparison of co-occurrence to sequence-based embedding methods. With MCA, although practitioners often retain two dimensions to visualise relationships, we demonstrated here that this fails to explain a substantial amount of known disease associations when using a large set of LTCs. We trialled a range of popular word embedding methods and given the applications of these methods in healthcare data are still relatively new, hypothesised that optimal hyperparameters for text data might not be optimal for disease code sequences, which do not follow the same syntactic relationships. We therefore experimented with a range of hyperparameters and found optimal ranges outside of the default values for standard text applications (see Supplementary Tables 3–6). When using unique code occurrences, both GloVe and SG performed similarly to MCA-30 in identifying known associations,

whilst CBOW's poorer performance was in line with previous reports in both text and healthcare data[37,59]. That NLP models and MCA produce similar results when using unique code sequences is unsurprising given the common basis in using disease-disease co-occurrence. Where NLP models showed improvement (see SG-M in Fig. 2) was when using longer sequences that included recurrent codes in the record, thus utilising additional information beyond direct co-occurrence. Previous studies have shown that sequence-based models have superior predictive performance for a range of outcomes[19,60], but we additionally found that the generated embeddings also better reflect clinically known disease associations. Applied to text, NLP methods such as GloVe and word2vec are effective at word analogy tasks, as compared with methods using co-occurrence alone. However, in a disease context, there are no clear equivalent disease analogy tasks, which might explain the relatively small improvement of SG-M over MCA.

More recent NLP architectures, including transformers such as BERT, make use of the full sequence of medical codes, and incorporate attention mechanisms that can retain longer-term dependencies often lost using methods such as word2vec[61]. Transformer architectures may also be used to generate word embeddings, but when applied to EHR data, require computationally intensive pre-training that was outside the scope of the current work[19,60]. Furthermore, the embeddings generate by BERT for a disease are dependent on the surrounding context (i.e., the other diseases which occur in sequence in a person) which adds complexity to their interpretation in a health context and to the identification of a single disease representation.

To help alleviate the lack of a gold-standard set of disease clusters, we created a list of established disease pairs and used these to compare across methods, finding MMCD to perform substantially better than both k-means and hierarchical clustering, particularly at finer resolutions. K-means and hierarchical clustering both produced unbalanced clusters with a large, dominant cluster and other smaller clusters of rarely occurring diseases,

likely due to the high dimensional and noisy nature of the disease representations, which is a well-characterised problem affecting many clustering algorithms[62]. The effect was more marked for MCA-30, suggesting a smoothing effect of the SG-M-generated embeddings when compared to MCA. However, in both cases, MMCD produced more balanced clusters, likely due to both the sparsification of the network using the MST-CkNN algorithm and the clustering cost function (Markov Stability), which enables MMCD to overcome such problems when using highly dimensional data.

The clustering of diseases is seen as a key part of elucidating the complexity of multimorbidity. Disease clusters may have an important role in clinical education, acting as heuristics for clinicians to prompt them to consider the co-existence of other diseases in a cluster, or proactive interventions to prevent their development. However, from a public health perspective, it remains open to research how disease clusters can best be applied, and whether they lead to better identification of shared risk factors or better prediction of clinical endpoints, compared with use of a person's individual diseases. For healthcare organisations, developing services that target clusters of co-occurring diseases might hypothetically help to minimise fragmentation of patient care, reducing the need for those with multiple conditions to see multiple specialists[63]. However, previous research has highlighted the vast number of unique disease combinations in the population[64] and further work is needed to understand the relationships between disease clusters and people, and whether clusters can reduce this complexity in a manner that is meaningful to health service design.

In future, we plan to extend our methods to cluster patients directly, using approaches analogous to topic modelling and document embeddings in NLP[65,66], specifically using large language models, such as BERT, which may provide additional insights into the similarity of disease sequences across people[21,60,67]. Although previous studies have evaluated the association of disease clusters with patient outcomes[10,68] we believe that evaluating outcomes should be reserved for clusters of patients, rather than clusters of diseases. Indeed, our preliminary assessment showed that disease clusters are not directly representative of patients, as relatively few patients were allocated to a single disease cluster, even when only two of their diseases were randomly sampled. Various approaches have been used to assign patients to disease clusters based, e.g., on a patient having one or more[69], two or more[70,71], or three or more[72] diseases in a cluster. However, these methods can assign patients to multiple clusters, an issue that will escalate with a larger number of clusters, and which can bias assessment of outcomes and complicate clinical use[73]. Alternatively, in a similar manner as can be applied to generate document vectors, patients could be represented by the sum or average of their disease vectors. However, it is unclear whether this approach is suitable when applied to representations of a person and is a focus for future research.

With MCA, age was a strong contributor to the first dimension which explained the largest proportion of variance. Furthermore, some of our clusters reflected sex differences, such as the clustering of gynaecological and breast malignancies at finer resolutions. Future work could consider stratification of clustering by age and sex, which may increase the ability to detect associations between less common diseases. Although our study already used a larger number of diseases (212) than previous studies of multimorbidity disease clustering, further increasing the number of conditions or using individual diagnostic codes (rather than categorising into diseases) may also increase the ability to detect novel associations at finer resolutions.

A strength of our study is the use of a large and representative sample of patients registered to general practices in England which enhances the generalisability of our results[27]. We used a larger set of LTCs than used before in multimorbidity research, making our findings more representative of disease patterns amongst patients. Although we experimented with a range of hyperparameters for our NLP algorithms, it is possible that better-performing models exist outside the range of hyperparameters tested. Hyperparameter optimisation could also be performed over our whole pipeline from generation of the disease representation to clustering. However, without a ground truth set of clusters, this might risk overfitting our clusters to represent the known disease associations, which would limit

exploratory findings. Although our best-performing SG-M model explained only 56% of the known disease pairs, the pairs were not generated to result in the assignment of all pairs within the top 10 nearest neighbours, and the number of nearest neighbours that would capture all the disease pairs is likely to vary according to disease. Indeed, when evaluating instead against a less strict top 20 nearest neighbours, the percentage of disease assignment increased to 71% (Supplementary Fig 4).

To compare embedding models, we developed a list of clinically-established disease associations. However, this list is not exhaustive, and may be biased towards inclusion of more common conditions which have a stronger evidence base. Furthermore, the combination of conditions included in the CALIBER study may also lead to bias in the embeddings and the clustering. For example, several unique conditions describe forms of liver disease and its sequelae (alcoholic liver disease, hepatic failure, cirrhosis, portal hypertension, oesophageal varices), which may all represent the same pathophysiological process, and so inflate similarity metrics between these conditions. This could explain the prominent separation of liver diseases on the second dimension in MCA (Supplementary Fig 3). However, other authors using different data sources and definitions have found similarly strong clustering of liver-related conditions[25]. Similarly, the stability of the metabolic cluster across resolutions in MCA-30 may in part stem from the inclusion of more diseases of this type in the code-lists (five diseases representing diabetes and its complications, and four representing cholesterol and triglycerides).

There are examples where a disease may be classified as both a specific and non-specific version of the same disease, both of which may appear in a patient's record. For example, codes for 'Diabetes: other or not specified' may be found in a patient's record in addition to those for 'Type 2 diabetes' or 'Type 1 diabetes', and similarly 'Stroke: not otherwise specified' in addition to 'Ischaemic stroke' or 'Intracerebral haemorrhage'. In these cases, the non-specific disease may represent use of generic codes used across each disease subtype, rather than that the disease itself is 'other' or unspecified' and are likely to be explained by clinician coding practices and the specificity of the available codes. As a result, the embeddings and clusters generated from routinely collected EHR data as used here reflect not only disease co-occurrence, but factors related to patients, clinicians, and healthcare organisations[56].

## Conclusion

In conclusion, using a representative cohort of over ten million people registered to general practices in England, we found clusters of diseases corresponding to both established and novel patterns. Clusters derived from co-occurrence-based embedding methods tended to be more straightforward to interpret than those from sequence-based NLP embedding methods, likely reflecting the additional relationships captured in disease sequences. Our multi-resolution approach highlights the nearly hierarchical structure of disease clusters but with notable exceptions that indicate the complexity of categorising certain diseases into a single set of inclusive clusters. Our study demonstrates the promise of these methods for identifying patterns of disease clusters within highly dimensional healthcare data, which could be used to facilitate discovery of associations between diseases in the future and help in optimising the management of people with multimorbidity, which is a priority for health systems globally.

## Data availability

The data that support the findings of this study are not openly available due to the risk of patient identification. Data can be requested from CPRD for users meeting certain requirements as described here: https://cprd.com/research-applications. Source data underlying Figs. 2 and 7 can be found in Supplementary Tables 9-11.

## Code availability

The code lists and embeddings generated from this work are available to download from: https://tbeaney.github.io/MMclustering/[74].

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

## Acknowledgements

This research is funded through a clinical PhD fellowship awarded to T.B. from the Wellcome Trust 4i programme at Imperial College London. Data management was provided by the Big Data and Analytical Unit (BDAU) at the Institute of Global Health Innovation (IGHI), Imperial College London. We thank Mark Cunningham for assistance with data extraction and management. We are grateful for the support of the NIHR Imperial Biomedical Research Centre. J.C. acknowledges support from the Wellcome Trust (215938/Z/19/Z). D.S. is supported by an Imperial College and National Institute of Health Research (NIHR) Post-Doctoral, Post-CCT research fellowship. T.W., A.M., and P.A. acknowledge support from the National Institute for Health and Care Research (NIHR) Applied Research Collaboration Northwest London. M.B. acknowledges support from EPSRC grant EP/N014529/1 supporting the EPSRC Centre for Mathematics of Precision Healthcare. Infrastructure support was provided by the NIHR Imperial Biomedical Research Centre. We thank Dominik Schindler for helpful discussions and help with the use of the PyGenStability package. The views expressed in this publication are those of the authors and not necessarily those of the NHS, the NIHR, the Wellcome Trust or the Department of Health and Social Care.

## Author contributions

T.B., P.A., and M.B. conceived the study, and T.B., J.C., and M.B. were involved in the study design and methodology. T.B., J.C., and D.S. developed the set of known disease associations. T.B. performed the data curation and formal analysis, with supervision from J.C. and M.B. T.B. wrote the first draught of the paper. T.B., J.C., D.S., T.W., A.M., P.A., and M.B. contributed to the interpretation of the results and the critical revision of the manuscript.

## Competing interests

The authors declare no competing interests.

## Ethical approval

Data access to the Clinical Practice Research Datalink (CPRD) and ethical approval was granted by CPRD's Research Data Governance Process on 28th April 2022 (Protocol reference: 22_001818). Patients are able to opt-out

of sharing data for research purposes and only anonymised data are provided as described here: https://www.cprd.com/safeguarding-patient-data.
