## [Peer Review File · Communications Medicine]

Reviewers' comments:

Reviewer #1 (Remarks to the Author):

#####

General overview

#####

We thank the editors and authors for the opportunity to review the study "Identifying multi-resolution clusters of diseases in ten million patients with multimorbidity in primary care in England" for publication in Nature Reviews Communications. The study focused used novel graph clustering methods (Markov Multiscale Community Detection - MMCD), in combination with latent embedding methods (MCA, word2vec, GloVe), to derive semantically plausible clusters of primary care patients sharing similar comorbid archetypes/patterns. The methods were illustrated using a large application dataset (UK CPRD primary care EMR dataset, consisting over >10M patients and their associated primary care clinical records). We agree with the authors primary hypothesis, i.e. that prevention/management of comorbid conditions in primary care is an important clinical challenge, that dichotmization of of comorbidity is an over-simplification relative to disease trajectories associated with particular comorbid conditions, and that novel clustering algorithms applied to primary care codified nomenclatures offers an opportunity for improved understanding of multi-morbidity. The study research problem is interesting and well-motivated; the dataset is large; the methods are well-described; and the inferences are plausible. Further, visualizations of results are excellent and aid intuitive understanding of study findings. As such, I recommend this paper for publication, with minor revisions.

#####

Minor comments for clarification

#####

Mapping from MedCodes to CALIBRE LTC Codes.

-- is this a heuristic mapping, derived by subject matter experts in primary care and specialist medicine?

-- i.e. if sets of medcodes share a reasonable semantic relationship, they are grouped together into 212 LTC codes? Kind of a semantic normalization step?

-- Perhaps describe over-arching CALIBRE vision in creating these smaller number of code-sets, what are guiding principles?

Additional baseline models:

- Bertopic applied to code sequences? Grootendorst et al. <https://github.com/MaartenGr/BERTopic>
- Advantage: kind of a cheap wrapper around more advanced embedding and clustering methods? May work well for disease code clustering? Easy software?
- Disadvantage, GPUs

Why not BERT and related LLM embeddings in this paper?

-- What are limiting factors or barriers to implementation?

** Transfer learning from English texts, to clinical code sequences (i.e. they may require extensive training, not fine-tuning and transfer learning for performance)

** Lack of access to GPUs on many primary care HPC clusters

** Time

Available software?

-- Major data modelling pipeline elements are gensim (word2vec/GloVe embeddings), STATA (MCA embeddings), and pygenstability (MMCD)

-- Scripts/Notebooks to illustrate computation associated with the inferential pipeline

HP tuning for NLP models

-- More exhasutive HPO over NLP embedding hyper-parameters?

-- Current approach more a grid-based evaluation over small number of hyper-parameters?

** Similarly, could the whole pipeline beenefit from HPO over NLP embedding, heuristics around inducing sparsity in graph, and MMCD clustering hyper-parameters?

** Random search, annealing, etc.

Representation of patient embeddings; from input code embeddings?

-- Are patients an average of their code-embeddings?

Reviewer #2 (Remarks to the Author):

Brief Summary

This paper explores the identification of clusters of co-occurring diseases within a population of 10 million patients in England. The authors have developed data-driven representations for 212 diseases and conducted two key experiments. The first compares disease representations using multiple correspondence analysis and word embeddings, focusing on cross-sectional and sequential representations. The second experiment compares methods for clustering diseases, namely multiscale graph-based clustering, K-means, and hierarchical clustering.

Overall Impression

The paper presents a comprehensive and detailed analysis of two significant problems in multimorbidity research using Electronic Health Records (EHRs): how should disease codes be represented and what types of clustering should be performed. Literature is abundant in methods that perform disease clustering, although only a few studies directly compare clustering techniques, and to the best of my knowledge, none compare different types of disease representation in this regard.

The discussion on word representation and the direct comparison of word embedding techniques for representing structured data, in this case, disease codes from EHRs, is extensive. The rationale for choosing specific hyperparameters, such as vector size and window size, which significantly influence results, is particularly well-articulated. The authors' approach of optimizing for known disease associations, defined by clinical knowledge, is insightful and deserves highlighting.

Regarding disease clustering, this work is strengthened by the size of the dataset and the use of a method that is underreported yet appears to offer superior results compared to typical clustering algorithms used in previous works. The direct comparison of different clustering techniques and results using various disease representations is notable. The authors discuss meaningful associations and acknowledge the limitations inherent in secondary data analysis, such as data entry bias.

Specific Comments and Recommendations:

1. Representation Techniques: The use of multiple correspondence analysis and various word embedding models (CBOW, skip-gram, GloVe) is well-executed and detailed. However, the manuscript could benefit from a more detailed explanation of the choice of these particular models and their relevance to disease representation.
2. Static vs. Sequential Representation of Disease Codes: This discussion is particularly valuable for the current state of the art in methods using structured data. The authors could elaborate more on this topic, including the impact of clinical history size on the effectiveness of sequential data representation.
3. Clustering Methods: The comparison between multiscale clustering, K-means, and hierarchical clustering is insightful. More context on why K-means and hierarchical clustering were less optimal, backed by additional references or examples, would be beneficial.
4. Data Interpretation: The findings on multimorbidity prevalence and the patterns observed in different clustering resolutions are intriguing. There could be more discussion on the practical implications of these findings, particularly in terms of public health strategy and disease management.

Conclusions

Overall, the paper is a valuable contribution to the field, especially with its innovative approach to disease clustering and data representation in EHRs. The authors effectively address the complexity of disease classification and the potential of data-driven approaches in understanding health patterns. The study's findings have significant implications for understanding multimorbidity and could inform future healthcare strategies.

Reviewer #3 (Remarks to the Author):

Major points

- By cosine similarity, the best performing embedding methods still only recapitulated approximately 55% of the curated set of known disease-disease associations. I think further discussion of the possible explanations and significance of this is required. Has this been tested using more contemporary approaches (e.g., a variational autoencoder, or deep learning-based NLP methods)? If so, how do these compare?
- The mechanism through which the optimal clustering solutions have been identified is not clear to me from the text. Linked to this, Figure 3 and Figure 4 (which purport to showing how the numbers of clusters was selected) would benefit from a more informative legend and/or footnote documenting what is contained within each subplot.
- The authors identify clusters that were deemed to arise due to the co-occurrence of disease codes within strata of age. Further discussion around this is required.
- Further contextualisation is required regarding the significance of the study findings, both at the clinical and organisational levels. How might these findings be used to benefit service organisation, for example (as is suggested in the introduction)?

Minor points:

- In the results subsection entitled 'Disease embeddings', the following statement is made: "Using this set of disease pairs, we retained 30 dimensions from MCA as the number that optimised known disease pairs being assigned in the top ten nearest neighbours to each disease based on the cosine similarity calculated from the MCA embeddings". I take this to mean that the 30 dimensions was selected as the hyperparameter of MCA that led to the optimal recapitulation of known disease pairs using this data driven approach. Is this correct? Please consider editing this sentence for greater clarity.
- I don't think figure 8 needs to be included in the main manuscript. This could be moved to a supplementary appendix.

Reviewer replies:

Reviewer 1

General overview

We thank the editors and authors for the opportunity to review the study "Identifying multi-resolution clusters of diseases in ten million patients with multimorbidity in primary care in England" for publication in Nature Reviews Communications. The study focused used novel graph clustering methods (Markov Multiscale Community Detection - MMCD), in combination with latent embedding methods (MCA, word2vec, GloVe), to derive semantically plausible clusters of primary care patients sharing similar comorbid archetypes/patterns. The methods were illustrated using a large application dataset (UK CPRD primary care EMR dataset, consisting over >10M patients and their associated primary care clinical records). We agree with the authors primary hypothesis, i.e. that prevention/management of comorbid conditions in primary care is an important clinical challenge, that dichotomization of of comorbidity is an over-simplification relative to disease trajectories associated with particular comorbid conditions, and that novel clustering algorithms applied to primary care codified nomenclatures offers an opportunity for improved understanding of multi-morbidity. The study research problem is interesting and well-motivated; the dataset is large; the methods are well-described; and the inferences are plausible. Further, visualizations of results are excellent and aid intuitive understanding of study findings. As such, I recommend this paper for publication, with minor revisions.

We thank the reviewer for their thorough review and constructive comments.

Minor comments for clarification

1. Mapping from MedCodes to CALIBRE LTC Codes.

-- is this a heuristic mapping, derived by subject matter experts in primary care and specialist medicine?

-- i.e. if sets of medcodes share a reasonable semantic relationship, they are grouped together into 212 LTC codes? Kind of a semantic normalization step?

-- Perhaps describe over-arching CALIBRE vision in creating these smaller number of code-sets, what are guiding principles?

Indeed, the mapping was derived by clinical consensus from the CALIBER study, which is publicly available via the HDRUK phenotype library, and represents a diverse set of conditions relevant to primary care. We agree that this is in effect semantic normalization, by grouping together codes which have a similar meaning into disease categories and essentially reducing the complexity of inputs. We have added further detail to the Methods ('Disease definitions' section) where we explain the process by which the codes were developed with reference to the CALIBER study, including references here.

2. Additional baseline models:

- Bertopic applied to code sequences? Grootendorst et al.
<https://github.com/MaartenGr/BERTopic>

- Advantage: kind of a cheap wrapper around more advanced embedding and clustering methods? May work well for disease code clustering? Easy software?
- Disadvantage, GPUs

Thank you for providing this link, and we agree this would be an interesting application to explore. However, it would require significant pre-training to first generate the word (disease) embeddings, given the differences in coded (structured) data to the original BERT models. The work in this manuscript is focussed on disease representations alone, and we have not gone down the route of topic modelling, which instead ascribes a distribution of topics over people. However, this is an interesting research direction which we propose to follow up by exploring patient-level representations. This would allow us to compare topic modelling and transformer-derived embeddings, and how these perform differently for prediction.

We have alluded to this in the first paragraph of 'Future work' in Discussion.

3. Why not BERT and related LLM embeddings in this paper?
- What are limiting factors or barriers to implementation?
 - ** Transfer learning from English texts, to clinical code sequences (i.e. they may require extensive training, not fine-tuning and transfer learning for performance)
 - ** Lack of access to GPUs on many primary care HPC clusters
 - ** Time

We did consider including those here, but BERT and other LLMs require significant pre-training specific to EHR data – even existing 'off-the-shelf' models for EHR data such as Med-BERT and BEHRT require pre-training to the specific dataset. As commented on above, we will consider this as a future area of research where we will consider different BERT architectures for creating patient-level representations. Given the already large number of comparisons in the current manuscript, we felt this was out of scope here.

In addition to the complexities of extensive pre-training, an additional complexity of BERT architectures is how to interpret the context-dependence of the word/disease embeddings. For example, 'Dementia' may have a different embedding in isolation, than in the context of a sequence of 'Hypertension Stroke Dementia', and it is not clear which of these is the 'correct' representation, as many diseases (e.g., secondary malignancies) are highly unlikely to occur in isolation. This makes the comparison of embeddings more complex and out of the scope of the current paper, but an important area for future work!

We have added a new second paragraph to the Discussion, 'Implications for embedding and clustering methods' to discuss these points.

4. Available software?
- Major data modelling pipeline elements are gensim (word2vec/GloVe embeddings), STATA (MCA embeddings), and pygenstability (MMCD)

-- Scripts/Notebooks to illustrate computation associated with the inferential pipeline

We thank the reviewer for this suggestion. We will make our notebooks available on our GitHub repository, including the MMCD codes we used. We have also made the disease representations available for download (for both MCA and SG algorithms).

5. HP tuning for NLP models

-- More exhasutive HPO over NLP embedding hyper-parameters?

-- Current approach more a grid-based evaluation over small number of hyper-parameters?

** Similarly, could the whole pipeline beenfit from HPO over NLP embedding, heuristics around inducing sparsity in graph, and MMCD clustering hyper-parameters?

** Random search, annealing, etc.

Although our hyperparameter grid-based search space was relatively confined around default settings, we found that there was very little change in the parameters. Nevertheless, we agree that a more extensive hyper-parameter optimisation could lead to better performing models subject to additional computation time. We were also conscious about the risk of 'overfitting' to our pre-defined list of associations. Likewise, when considering optimisation of the whole pipeline, we might risk identifying clusters which fitted our expectations. This would limit the potential for identifying novel connections. We therefore believe that our approach represents a pragmatic balance between broadly optimising the disease representations to a small set of known associations, while still allowing freedom for unsupervised clustering of the representations.

We have added a sentence to the first paragraph of 'Strengths and Limitations' to acknowledge this issue.

6. Representation of patient embeddings; from input code embeddings?

-- Are patients an average of their code-embeddings?

This is an interesting question and something we will investigate in future work. Although when dealing with text the average of word embeddings might be a good representation of a document, this approach might not lead to a good representation of a patient, particularly where repeated codes are included (code frequency in the EHR is not in itself an objective marker of health status, as we showed in other work: <https://bmjopen.bmj.com/content/13/9/e072884>).

In the current manuscript, we have thus focussed on disease representations and disease clusters, which we believe are quite different in terms of their assumptions than approaches which directly generate patient representations/clusters. This direction of research will be studied in future work.

We have added two sentences to the first paragraph of 'Future work' to discuss these issues.

Reviewer 2

Brief Summary

This paper explores the identification of clusters of co-occurring diseases within a population of 10 million patients in England. The authors have developed data-driven representations for 212 diseases and conducted two key experiments. The first compares disease representations using multiple correspondence analysis and word embeddings, focusing on cross-sectional and sequential representations. The second experiment compares methods for clustering diseases, namely multiscale graph-based clustering, K-means, and hierarchical clustering.

Overall Impression

The paper presents a comprehensive and detailed analysis of two significant problems in multimorbidity research using Electronic Health Records (EHRs): how should disease codes be represented and what types of clustering should be performed. Literature is abundant in methods that perform disease clustering, although only a few studies directly compare clustering techniques, and to the best of my knowledge, none compare different types of disease representation in this regard.

The discussion on word representation and the direct comparison of word embedding techniques for representing structured data, in this case, disease codes from EHRs, is extensive. The rationale for choosing specific hyperparameters, such as vector size and window size, which significantly influence results, is particularly well-articulated. The authors' approach of optimizing for known disease associations, defined by clinical knowledge, is insightful and deserves highlighting.

Regarding disease clustering, this work is strengthened by the size of the dataset and the use of a method that is underreported yet appears to offer superior results compared to typical clustering algorithms used in previous works. The direct comparison of different clustering techniques and results using various disease representations is notable. The authors discuss meaningful associations and acknowledge the limitations inherent in secondary data analysis, such as data entry bias.

We thank the reviewer for their appreciation of our work and helpful comments for improving the article.

Specific Comments and Recommendations:

1. Representation Techniques: The use of multiple correspondence analysis and various word embedding models (CBOW, skip-gram, GloVe) is well-executed and detailed. However, the manuscript could benefit from a more detailed explanation of the choice of these particular models and their relevance to disease representation.

We thank the reviewer for this suggestion. We have added additional text into the Methods, 'Generating disease embeddings' section, where we highlight a key difference of word2vec and GloVe over co-occurrence methods, in explaining word analogies. We have also added to the first paragraph of 'Implications for embedding and clustering methods' in Discussion to highlight the key difference between co-occurrence and word2vec/GloVe models and performance in text applications, and

why these may differ applied to diseases, where obvious word analogy tasks do not exist. For instance, unlike the case of 'king is to queen as man is to woman', we are not aware of any equivalent analogies that can be applied to a disease context (we spent some time trying to think of some!).

2. Static vs. Sequential Representation of Disease Codes: This discussion is particularly valuable for the current state of the art in methods using structured data. The authors could elaborate more on this topic, including the impact of clinical history size on the effectiveness of sequential data representation.

We agree this is an interesting topic, and are planning future work that explores this in more depth. We have added further details to the second paragraph of 'Implications for embedding and clustering methods' (in Discussion), where we discuss how methods that capture full sequences such as BERT could also be used, but have highlighted the complexity in interpreting the contextual representations of diseases versus a static, context independent representation as given by word2vec/GloVe.

3. Clustering Methods: The comparison between multiscale clustering, K-means, and hierarchical clustering is insightful. More context on why K-means and hierarchical clustering were less optimal, backed by additional references or examples, would be beneficial.

Both k-means and hierarchical clustering are known to be sensitive to noise and high dimensional data, and can lead to unbalanced clusters. We have added text, along with a reference giving an overview of these problems, to the third paragraph of 'Implications for embedding and clustering methods.'

4. Data Interpretation: The findings on multimorbidity prevalence and the patterns observed in different clustering resolutions are intriguing. There could be more discussion on the practical implications of these findings, particularly in terms of public health strategy and disease management.

We have added a new section to the Discussion on 'Implications for public health and multimorbidity research', where we briefly cover some of the implications. We believe there remains a need for further research to identify how disease clusters can be used to understand risk factors or predict adverse health outcomes and we remark there on these issues.

Conclusions

Overall, the paper is a valuable contribution to the field, especially with its innovative approach to disease clustering and data representation in EHRs. The authors effectively address the complexity of disease classification and the potential of data-driven approaches in understanding health patterns. The study's findings have significant implications for understanding multimorbidity and could inform future healthcare strategies.

We thank the Reviewer for their assessment of our work.

Reviewer 3

Major points

- By cosine similarity, the best performing embedding methods still only recapitulated approximately 55% of the curated set of known disease-disease associations. I think further discussion of the possible explanations and significance of this is required. Has this been tested using more contemporary approaches (e.g., a variational autoencoder, or deep learning-based NLP methods)? If so, how do these compare?

As noted in our response to Reviewer 1, we did consider more state-of-the-art models, such as BERT, but these bring additional challenges with the interpretation of context-dependence of the disease embeddings, and significant computational training, which we have now discussed at the end of the second paragraph of 'Implications for embedding and clustering methods'.

We would not necessarily expect all disease pairs to be assigned, and the value of N in the top N most similar diseases which captures all the disease pairs is likely to vary according to disease. Use of the top 10 is a relatively strict criterion (approximately the top 5% of conditions) and when increasing from 10 to 20, 71% of pairs are captured in the top 20 nearest neighbours (Supplementary Figure 4). We have now added an explicit acknowledgement of these points at the end of the first paragraph of 'Strengths and limitations' in the Discussion.

- The mechanism through which the optimal clustering solutions have been identified is not clear to me from the text. Linked to this, Figure 3 and Figure 4 (which purport to showing how the numbers of clusters was selected) would benefit from a more informative legend and/or footnote documenting what is contained within each subplot.

We thank the reviewer for pointing this out, and we apologise for the lack of clarity. We have amended the text to the caption to better explain the separate subplots and to explain the differences in the NVI and block NVI.

- The authors identify clusters that were deemed to arise due to the co-occurrence of disease codes within strata of age. Further discussion around this is required.

We have added additional text to the first paragraph of 'Clinical implications' to discuss the findings of age, and also highlight the different attributes by which conditions might be clustered.

- Further contextualisation is required regarding the significance of the study findings, both at the clinical and organisational levels. How might these findings be used to benefit service organisation, for example (as is suggested in the introduction)?

We have added a section on 'Implications for public health and multimorbidity research' where we discuss these findings, and the need for further research into how disease clusters can help identify shared risk factors or common health outcomes in people.

Minor points:

- In the results subsection entitled 'Disease embeddings', the following statement is made: "Using this set of disease pairs, we retained 30 dimensions from MCA as the number that optimised known disease pairs being assigned in the top ten nearest neighbours to each disease based on the cosine similarity calculated from the MCA embeddings". I take this to mean that the 30 dimensions was selected as the hyperparameter of MCA that led to the optimal recapitulation of known disease pairs using this data driven approach. Is this correct? Please consider editing this sentence for greater clarity.

Thank you. We agree this was unclear, and have rephrased for clarity.

- I don't think figure 8 needs to be included in the main manuscript. This could be moved to a supplementary appendix.

We think Figure 8 is still of interest as a comparison of the MCA embeddings to the SG-M embeddings and have retained this, particularly given co-occurrence methods are more standard within epidemiological literature on multimorbidity.

REVIEWERS' COMMENTS:

Reviewer #1 (Remarks to the Author):

The authors have adequately addressed the comments raised during my revision. At this stage, I suggest the research study be published by the journal with no additional revisions.

Reviewer #2 (Remarks to the Author):

I am happy with the changes made by the authors concerning my points. I have nothing more to add.

Reviewer #4 (Remarks to the Author):

Many thanks for the invitation to review this revised manuscript looking to identify multi-resolutional clusters in a large primary care dataset from the UK.

I have been asked to review the authors rebuttal to reviewer #3's comments and revisions made to the manuscript.

The authors have provided detailed and satisfactory responses to this the reviewers comments. They have demonstrated a novel and highly topical method and present a clear future direction for how this approach may help utilise healthcare data to identify new associations between disease in multi-morbid patients.